# Development of SARS-CoV-2 Vaccine: Challenges and Prospects

**DOI:** 10.3390/diseases11020064

**Published:** 2023-04-20

**Authors:** Tooba Mahboob, Amni Adilah Ismail, Muhammad Raza Shah, Mohammed Rahmatullah, Alok K. Paul, Maria de Lourdes Pereira, Christophe Wiart, Polrat Wilairatana, Mogana Rajagopal, Karma G. Dolma, Veeranoot Nissapatorn

**Affiliations:** 1Faculty of Pharmaceutical Sciences, UCSI University, Kuala Lumpur 56000, Malaysia; 2Department of Medical Microbiology, Faculty of Medicine, University of Malaya, Kuala Lumpur 50603, Malaysia; 3H.E.J. Research Institute of Chemistry, International Center for Chemical and Biological Sciences, University of Karachi, Karachi 75270, Pakistan; 4Department of Biotechnology & Genetic Engineering, University of Development Alternative, Lalmatia, Dhaka 1209, Bangladesh; 5School of Pharmacy and Pharmacology, University of Tasmania, Hobart, TAS 7005, Australia; 6CICECO—Aveiro Institute of Materials & Department of Medical Sciences, University of Aveiro, 3810-193 Aveiro, Portugal; 7Institute for Tropical Biology and Conservation, University Malaysia, Sabah 88400, Malaysia; 8Department of Clinical Tropical Medicine, Faculty of Tropical Medicine, Mahidol University, Bangkok 10400, Thailand; 9Department of Microbiology, Sikkim Manipal Institute of Medical Sciences, Sikkim Manipal University, Gangtok 737102, Sikkim, India; 10School of Allied Health Sciences and World Union for Herbal Drug Discovery (WUHeDD), Walailak University, Nakhon Si Thammarat 80160, Thailand

**Keywords:** COVID-19, SARS-CoV-2, vaccines, variants

## Abstract

The WHO declared coronavirus disease 2019 (COVID-19) a pandemic in March 2020, which was caused by novel coronavirus severe acute respiratory coronavirus 2 (SARS-CoV-2). SARS-CoV-2 made its first entry into the world in November 2019, and the first case was detected in Wuhan, China. Mutations in the SARS-CoV-2 genome distressed life in almost every discipline by the extended production of novel viral variants. In this article, authorized SARS-CoV-2 vaccines including mRNA vaccines, DNA vaccines, subunit vaccines, inactivated virus vaccines, viral vector vaccine, live attenuated virus vaccines and mix and match vaccines will be discussed based on their mechanism, administration, storage, stability, safety and efficacy. The information was collected from various journals via electronic searches including PubMed, Science Direct, Google Scholar and the WHO platform. This review article includes a brief summary on the pathophysiology, epidemiology, mutant variants and management strategies related to COVID-19. Due to the continuous production and unsatisfactory understanding of novel variants of SARS-CoV-2, it is important to design an effective vaccine along with long-lasting protection against variant strains by eliminating the gaps through practical and theoretical knowledge. Consequently, it is mandatory to update the literature through previous and ongoing trials of vaccines tested among various ethnicities and age groups to gain a better insight into management strategies and combat complications associated with upcoming novel variants of SARS-CoV-2.

## 1. Introduction

### 1.1. Genome Structure and Characteristic of COVID-19

The causative agent of the current global pandemic also known as coronavirus disease (COVID-19) is severe acute respiratory syndrome coronavirus 2 (SARS-CoV-2), which was previously known as novel coronavirus (2019-nCoV). As shown in Figure 1, SARS-CoV-2 is an enveloped ribonucleic acid (RNA) virus packaged by nucleocapsid phosphoproteins that is diversely found in humans and wildlife. Coronavirus is known as one of the largest RNA viruses (26–32 kb) [1]. The genome of SARS-CoV consists of 14 open reading frames (ORFs) which encodes 27 structural and non-structural proteins. The primary structural proteins of coronavirus are membrane (M), spike (S) and nucleocapsid (N) proteins [2]. A total of seven species have been identified as moderately pathogenic to humans which includes alphacoronaviruses HCoV-229E and HCoV-NL63 and betacoronaviruses HCoV-OC43, HCoV-HKU1, SARS-CoV, MERS-CoV and SARS-CoV-2 [3]. They are known to infect the neurological, respiratory, enteric, and hepatic systems [1,2,3,4].

### 1.2. Epidemiology of SARS-CoV-2

The past few decades have seen endemic coronavirus outbreaks in the Middle East (2012) and China (2002). Yet again, we see the emergence of another outbreak due to a new strain called the SARS-CoV-2 virus, which is highly contagious [5]. The most recent outbreak initially presented as pneumonia of unknown etiology in a cluster of patients in Wuhan, China [6]. The repeated emergence of coronavirus points toward the animal-to-human and human-to-human transmission of this virus and the evolution of new coronaviruses in the future.

Since the 12 December 2019, Wuhan, China has been the epicenter of coronavirus activity [7,8]. It was first reported as an acute respiratory illness that nobody had heard of before. Some studies have suggested that bats may be the likely source of the SARS-CoV-2 virus. However, there is no evidence to suggest that the SARS-CoV-2 virus originated in a seafood market. This theory remains unproven. To a lesser extent, bats serve as a natural reservoir for a wide variety of coronaviruses, including viruses that are similar to MERS-Co-V and SARS-Co-V [9,10,11]. SARS-CoV-2 was anticipated to spread mostly through direct contact with wild animals through ingestion or through intermediary host species. On the other hand, neither the route(s) of transmission nor the source(s) of SARS-CoV-2 have been definitively established.

### 1.3. Mutant Variant of SARS-CoV-2

After the outbreak of COVID-19 in late 2019, the virus remained genetically stable for approximately 11 months. Subsequently, in late 2020, COVID-19 development has been characterized by the emerging of clusters of mutations of SARS-CoV-2 known as ‘variants of concern’ (VOC), which affect viral features that include virus transmissibility and antigenicity. It emerged most likely in response to the shifting of the human population immunological profile. Furthermore, Harvey and colleagues have extensively discussed SARS-CoV-2 mutant variants, particularly the spike protein (the main antigen) [12]. It was reported that most of the mutation found in genomes from circulating SARS-CoV-2 virions are likely to be neutral or modestly deleterious. This is because ‘neutral’ amino acid alterations, which have no apparent influence on the virus’s efficiency or adaptability, are much more common than the rare ‘high-effect’ mutations that are necessary for adaptation.

For instance, the spike protein amino acid change D614G was shown to be occurring more often and many times in the global SARS-CoV-2 population in April 2020 [13,14]. Additionally, the high dN/dS ratio in the coding sequence suggests a positive selection at codon position 614 [13,14]. This was further supported by subsequent investigations which revealed that D614G has improved infectivity [15,16] and transmission [17]. In November 2021, the SARS-CoV-2 variant known as Omicron (B.1.1.529) was discovered in Botswana, South Africa, and Hong Kong, and it has since been a cause for alarm as VOC. Concerns regarding Omicron’s possible high transmissibility stem from the fact that it has been detected in at least 85 countries, as indicated by SARS-CoV-2 genome data published on GISAID (https://www.gisaid.org), accessed on 18 October 2022. The multitude of mutations in the S protein, some of which are shared with past VOCs and others that are novel, is the most alarming component of Omicron. According to preliminary investigations, the ability of Omicron to elude humoral immune responses is quite high, as seen by the rapid decline in the neutralizing effectiveness of infection- and vaccine-elicited antibodies and sera against Omicron [18,19,20,21,22]. Nevertheless, it is unclear how well Omicron can elude cellular immunological responses, which is the other component of the adaptive immune system mediated by T cells. Table 1 showed the circulating VOCs and Omicron subvariants under monitoring (as of 12 October 2022) published on the WHO (https://www.who.int/activities/tracking-SARS-CoV-2-variants), accessed on 18 October 2022.

Most importantly, a better knowledge of the phenotypic effects of mutations across the SARS-CoV-2 genome and their implications for variant survivability may aid in elucidating the mechanisms of virus transmission and evolutionary success.

### 1.4. Clinical Pathology of COVID-19

Clinical manifestations associated with COVID-19 present a broad spectrum which include asymptomatic patients to multiorgan dysfunction. COVID-19 can be categorized based on severity of infection such as mild, moderate, severe and critical. The common symptoms of COVID-19-infected patients could be ordered as major and minor symptoms. Major symptoms include fever (98.6%), fatigue (69.6%), dry cough and body ache, whereby minor symptoms comprise dyspnea, gastrointestinal disorders headaches and skin lesions [23]. Rare cases of vasculitis-type skin eruption in association with COVD-19 have been reported in patients infected with SARS-CoV-2 [24]. Overall, the spectrum of symptoms was found to be varied continuously in respect to different variants. For instance, the Omicron variant of SARS-CoV-2, a highly contagious variant, caused less severe disease with lower replication in lung parenchyma in comparison with the previous variants. In a study conducted by Boscolo-Rizzo and team, a significant reduction has been observed in self-reported chemosensory dysfunction such as taste and smell when comparing patients infected with the Omicron variant and previous variants including Alpha, Beta, and Delta [25]. A multiorgan inflammatory syndrome associated with SARS-CoV-2 has been observed in older children linked with abdominal pain, cardiac dysfunction and shock having resemblance with Kawasaki disease [26]. Patients infected with SARS-CoV-2 may have minimal to no symptoms related to acute respiratory distress syndrome (ARDS) which can result into multiorgan failure. Additionally, the severity of symptoms depends on the age of patient and is probably associated with a weaker immune system. The most common comorbidities related to elderly patients infected with SARS-CoV-2 include hypertension, neurological disorders, diabetes, and cardiovascular diseases [25].

### 1.5. Immunopathology of COVID-19

As a member of the cytopathic virus family, during the replication, SARS-CoV-2 damages the host tissues by targeting cells [27]. Angiotensin-Converting Enzyme 2 (ACE2) is regarded as the primary receptor which enables the entry of SARS-CoV-2 into host cells. ACE2 receptors are commonly present on the surface of alveolar epithelial cells (Type II), which are vital for the gaseous exchange in lungs. ACE2 is also expressed on endothelial cells, type I pneumocytes and outside pulmonary tissues such as intestine, blood vessels, heart, urinary bladder, and kidney [28,29]. Patients infected with SARS-CoV-2 have reported a greater level of pro-inflammatory cytokines: mainly Interleukin-6 (IL-6), Interleukin-8 (IL-8), Interleukin-1β (IL-1β), Tumor Necrosis Factor Alpha (TNF-α), Interferon-inducible protein 10 (IP10), Monocyte chemoattractant protein-1 (MCP-1), and regulated upon activation, normal T cell expressed and secreted (RANTES) similar to MERS virus. Monocytes and macrophages are activated by the release of cytokines and chemokines, which resulted in the generation of cytokines to prime T cell adaptive immune response [30]. Most of the time, the activation of macrophages and monocytes will invade the virus and results in reduced inflammation. Whereby, in a few cases, an adverse inflammatory response associated with a storm of cytokines has been observed where patients have previous underlying medical conditions or immunocompromised immune systems. This storm of cytokines is commonly classified by elevated plasma cytokines levels including Interleukin-2 (IL-2), Interleukin-7 (IL-7), Interleukin-10 (IL-10), Granulocyte colony-stimulating factor (GCSF), IP-10, MCP-1, MIP-1α, and TNF-α [29,30,31,32]. Such pulmonary inflammation is uncontrolled and most likely causes death. Moreover, thrombosis and pulmonary embolism have also been linked with the severe infections of SARS-CoV-2. In this case, platelets are hyperactivated, which accumulates platelets at thrombin suboptimal concentrations probably due to the higher production of a clotting factor by the liver caused by inflammatory cytokines [33]. Similarly, it has been reported that the receipt of angiotensin receptor blockers (ARB) increases the risk of SARS-CoV-2 infections particularly in close communities [34]. Whereby, in another retrospective study, the use of ARBs did not present any significant affiliation with the severity or mortality or elevation in the diagnosis of COVID-19 [35]. Nevertheless, the mechanism behind the immune response and associated molecular pathways to SARS-CoV-2 are still vivid and vague. It is very important to clarify these pathways to develop an effective therapeutic agent especially for upcoming new variants of SARS-CoV-2.

### 1.6. Management Strategies of COVID-19

Isolation remained the best measure for the containment of this virus in the initial days of the SARS-CoV-2 pandemic [36]. Social distancing, the use of sanitizers and quarantine were a few of the measures to combat infections caused by SARS-CoV-2. In the early days of the coronavirus pandemic, there was no specific drug or vaccine available to treat the viral infection. Prior to the approval of the SARS-CoV-2 vaccination by the WHO, supportive management was essentially required for patients with a mild infection by utilizing acetaminophen, external cooling, and oxygen therapy nutritional supplements. On the contrary, critical patients required high-flow oxygen, glucocorticoid therapy, and convalescent plasma [1,37]. Currently, several vaccines and antiviral drugs have been approved to treat and reduce severity related to SARS-CoV-2 infections [37]. Recently, the Food and Drug Administration (FDA) approved monoclonal antibodies (mAbs) tixagevimab and cilgavimab for the pre-exposure prophylaxis of COVID-19 in individuals having severe allergies to SARS-CoV-2 vaccines. These monoclonal antibodies are injected via the intramuscular (IM) or intravenous (IV) route. Tixagevimab and cilgavimab exhibited potential activity against the Delta variant of SARS-CoV-2 by the inhibition of viral entry into the host [38]. IL-6 is a pro-inflammatory cytokine, and anti-IL6 drugs have been approved by the FDA for COVID-19 patients having systemic inflammation or respiratory failure, as these conditions could be associated with an elevated release of cytokine. There are two classes of IL-6 inhibitors approved by the FDA after examining in patients with systemic inflammation: anti-IL-6 receptor mAbs (e.g., tocilizumab, sarilumab) and anti-IL-6 mAbs (i.e., siltuximab) [39]. Furthermore, oral antivirals drugs are also available including molnupiravir and the combination of nirmatrelvir and ritonavir to treat COVID-19. These antivirals demonstrated a significant reduction in hospitalization, death, and severity of disease particularly among vaccinated individuals having compelling comorbidities in a real-life retrospective study [40]. The successful use of a member of the corticosteroid family, Dexamethasone, was also reported to reduced mortality in COVID-19 patients having intensive respiratory response [41]. A nucleotide inhibitor, Remdesivir, which has been approved by the FDA, was also reported to hinder the severity of SARS-CoV-2 infection with a low risk of hospitalization or death [42]. A similar outcome was reported by Piccicacco and co-workers based on the use of remdesivir and sotrovimab in an outpatient setting to reduce the risk of hospitalization [43].

This review gives a brief overview of the characteristics of SARS-CoV-2, the pathophysiology of the infection, and an update on the vaccines developed as shown in Figure 2 and Table 1 and Table 2.

**Table 1 diseases-11-00064-t001:** Circulating variants of concerns (VOCs).

	WHO Name	Pango Lineage	GISAID Clade	Next Strain Clade	Country/Date of First Detection	Date of Designation	Pathophysiology	Epidemiology
**Previously circulating VOCs**	Alpha	B.1.1.7	GRY	20I (V1)	United Kingdom	VOC: 18 December 2020	- nine mutations on the SARS-CoV-2 spike protein: two deletions and seven amino acid substitutions [44] - higher transmissibility (43–82% more transmissible) [44] - higher viral load [44] - longer duration of infection [44] - higher hospitalization rate [44] - higher mortality rate [44] - have higher reproduction numbers—a rate 40–90% higher than D614G - reinfection is lower than other strains [44] - susceptible to neutralizing antibodies by spike vaccines [44]	- as of February 2021, responsible for nearly 95% of SARS-CoV-2 transmission in England [44] - affects mainly healthier and younger patients [44] - variable reports on severity, some studies have shown it does not increase the severity and has been associated with milder disease [44]
					September 2020			
						Previous VOC: 9 March 2022		
	Beta	B.1.351	GH/501Y.V2	20H (V2)	South Africa	VOC: 18 December 2020	- 18 amino acid mutations (7 in the spike protein) and 3 amino acid deletions in the spike protein [44] - Mutations at the RBD: N501Y, E484K, and K417N [44] - higher transmissibility [44] - higher viral load [44] - higher reinfection [44] - higher vaccine escape [44] - higher hospitalization rate [44] - higher mortality rate [44] - more resistant to neutralization by sera from vaccinated individuals and convalescent plasma [44]	- As of January 2021, the variant had spread to several countries [44] - cases involving those <60 years old were found to have a higher hospitalization rate and admission rate to intensive unit care [44]
					May 2020			
						Previous VOC: 9 March 2022		
	Gamma	P.1	GR/501Y.V3	20J (V3)	Brazil,	VOC: 11 January 2021	- multiple spike protein mutations, including K417T, E484K, N501Y in the RBD; L18F, T20N, P26S, D138Y, R190S in the NTD; D614G and H655Y at C terminus in S1; and V1176F and T1027I in S2 [44] - higher transmissibility [44] - has a probability of reinfection of 6.4% [44] - higher viral load [44] - higher hospitalization [44] - resistant to neutralization [44]	- Associated with travel-related cases detected in Japan and São Paulo [44] - a higher hospitalization rate and admission rate to an intensive unit care especially for those <60 years of age - in terms of the disease severity, limited published data available [44]
					November 2020			
						Previous VOC: 9 March 2022		
	Delta	B.1.617.2	G/478K.V1	21A, 21I, 21J	India	VOI: 4 April 2021	- spike mutations: D614G, D950N, L452R, T19R, T478K, P681R, 156–157 deletion and R158G and G142D. The G142D is found only in some B.1.617.2; B.1.617.3, with the next strain name 20A, has spike protein mutations: D614G, D950N, L452R, P681R, T19R, E484Q, and G142D [44] - 60% more transmissible than B.1.1.7 - higher risk of hospitalization [44] - higher viral load [44] - increase severity [44] - higher vaccine escape [44]	- associated with more pediatric cases [44]
					October 2020	VOC: 11 May 2021		
						Previous VOC: 7 June 2022		
**Currently circulating VOCs**	Omicron ^1^	B.1.1.529	GR/484A	21K, 21L, 21M, 22A,	Multiple countries	VUM: 24 November 2021	- highly infectious, highly mutated, highly transmissible, and highly resistant to available vaccines [45] - several mutations in (or near) RBD, NTD, RBM, S2 domains and furin cleavage site, affecting antibody binding and ACE2 binding [45] - interacts less efficiently with neutralizing convalescent mAb [45]	- in early and mid-November 2021, it was detected in Botswana and several areas of South Africa, particularly Gauteng province [45]. Then, it became a global concern as nearly 150 countries had a surge in Omicron cases (including the USA, UK, Australia, France, Germany, Denmark, Japan, Netherlands, India, and other countries) [45]
				22B, 22C, 22D	November 21	VOC: 26 November 2021		

^1^ Includes BA.1, BA.2, BA.3, BA.4, BA.5 and descendent lineages. It also includes BA.1/BA.2 circulating recombinant forms such as XE.

**Table 3 diseases-11-00064-t003:** SARS-CoV-2 Vaccines.

Vaccine	Manufacturer/Country	Type	Antigen	Dose/Dosage	Efficacy	Overall Efficacy	Approved Countries	References
mRNA-1273/SpikeVax	Moderna (US)	mRNA	Full-length spike (S) protein with proline substitutions	100 μg: 2 doses (28 days apart)	100% 14 days after second dose	92.1% 14 days after first dose; 94.1% 14 days after second dose	EUA: the US, EU, Canada, and UK	[57,58,59,60]
BNT162b2/Comirnaty	Pfizer.BioNTech (US)	mRNA	Full-length S protein with proline substitutions	30 μg: 2 doses (21 days apart)	88.9% after 1 dose	52% after first dose; 94.6% 7 days after second dose	EUA: the US, EU, Canada, and UK	[61]
Ad26.CoV2.S	Janssen/Johnson & Johnson (US)	Viral vector	Recombinant, replication incompetent human adenovirus serotype 26 vector encoding a full-length, stabilized SARS-CoV-2 S protein	5 × 10^10^ viral particles: 1 dose	85% after 28 days; 100% after 49 days	72% in the US; 66% in Latin America; 57% in South Africa (at 28 days)	EUA: the US, EU, and Canada	[57]
ChAdOx1(AZS1222)/ Covishield	AstraZeneca/Oxford (UK)	Viral vector	Replication-deficient chimpanzee adenoviral vector with the SARS-CoV-2 S protein	5 × 10^10^ viral particles: 2 doses (28 days apart)	100% 21 days after first dose	64.1% after first dose; 70.4% 14 days after second dose	EUA: WHO/Covax, the UK, India, and Mexico	[57]
NVX-CoV2373/Nuvaxovid	Novavax, Inc (US)	Protein subunit	Recombinant full-length, prefusion S protein	5 μg of protein and 50 μg of Matrix-M adjuvant: 2 doses	Unknown	89.3% in the UK (after 2 doses); 60% in South Africa	EUA application planned	[62]
CVnCov	CureVa/GlaxoSmithKline (Germany)	mRNA	Prefusion stabilized full-length S protein of the SARS-CoV-2 virus	12 μg: 2 doses (28 days apart)	Unknown	Ongoing Phase 3 trial	Not applicable	[62]
Gam-COVID-Vac (Sputnik V)	Gamaleya National Research Center for Epidemiology and Microbiology (Russia)	Viral vector	Full-length SARS-CoV-2 virus glycoprotein S carried by adenoviral vectors	10^11^ virus particles per dose for each recombinant adenovirus: 2 doses (first rAd26; second rAd5) (21 days apart)	100% 21 days after first dose	87.6% 14 days after first dose; 91.1% 7 days after second dose	EUA: Russia, Belarus, Argentina, Serbia, UAE, Algeria, Palestine, and Egypt	[63]
CoronaVac	Sinovac Biotech (China)	Inactivated virus	Inactivated CN02 Strain of SARS-CoV-2 created from Vero cells	3 μg with aluminum hydroxide adjuvant: 2 doses (14 days apart)	Unknown	Phase 3 data not published; reported efficacy 14 days after dose 2: 50.38% (mild) and 78% (mild to severe) in Brazil, 65% in Indonesia, and 91.25% in Turkey	EUA: China, Brazil, Columbia, Bolivia, Brazil, Chile, Uruguay, Turkey, Indonesia and Azerbaijan	[62]
BBIBP-CorV	Sinopharm (China)	Inactivated virus	Inactivated HB02 strain of SARS-CoV-2 created from Vero cells	4 μg with aluminum hydroxide adjuvant: 2 doses (21 days apart)	Unknown	Phase 3 data not published; unpublished reported 79% and 86% efficacy	EUA: China, UAE, Bahrain, Serbia, Peru, and Zimbabwe	[57]
BBV152	Covaxin (Bharat Biotech, India)	Inactivated virus	Whole SARS-CoV-2 Virion (Strain: NIV-2020-770), inactivated Vero Cell	6 μg of whole-virion with aluminum hydroxide adjuvant: 2 doses (28 days apart)	78% after second dose	77.8% (symptomatic); 93.4% (severe); 63.6% (asymptomatic)	Asia, Europe, Africa, South America, North America, Oceania (Australia)	[64]
Ad5-nCoV-S recombinant (Ad5-nCoV)	CanSinoBio/Convidecia	Viral vector	Replication-defective adenovirus type 5 vector expressing the SARS-CoV-2 S protein	0.5 × 10^11^ virus particles per dose for each recombinant adenovirus: 1 dose	90.07% (severe) 28 days after single dose and 95.47% (severe) 14 days after single dose.	65.28% (symptomatic) 28 days after single dose vaccination, and 68.83% (symptomatic) 14 days after single dose.	Asia, Europe, and Latin America	[64]

## 2. Current Update on Various Vaccines Targeting SARS-CoV-2

### 2.1. mRNA Vaccines

The vaccine based on the messenger RNA (mRNA) platform served an exceptional role in management and specifically helped reduce the severity of infections associated with SARS-CoV-2: particularly Alpha, Beta and Delta variants. It showed its potential to be considered as a more focused platform for future vaccine development. It proved its great potential to significantly deal with vaccine growth targeting many pathogens. The rapid production of mRNA vaccines, demonstration of T helper 1 (Th1) and T helper 2 (Th2) response, mRNA modifications, stabilization, and delivery methods account for the recent success of mRNA vaccines [62]. RNA moieties are smaller and easily transcribed in vitro as compared to DNA. Lipid nanoparticles have been utilized in the delivery of mRNA vaccines to shield the prefusion-stabilized S protein-encoding mRNA. In addition, a range of other materials have been used in the development of vaccines [62]. Pfizer/BioNTech BNT162b2 and the Moderna mRNA-1273 are included among successful mRNA vaccines, which have been used in clinical trials with more than 90% effectivity against SARS-CoV-2 [57], whereby very limited side effects have been observed, which mainly include local and systemic reactogenicity. A total of 30 µg of Pfizer/BioNTech BNT162b2 and 100 µg Moderna mRNA-1273 vaccines are usually administered in two doses with a maximum gap of 21 and 28 days, respectively. Pfizer/BioNTech BNT162b2 and the Moderna mRNA-1273 vaccines have improved stability by reduction in undesirable type I interferon immune response [58,59,60]. Multiple studies have been planned to check the efficacy of mRNA vaccines in children and new variants of SARS-CoV-2. Storage condition for Moderna mRNA-1273 and Pfizer/BioNTech BNT162b2 vaccines are −25 to −15 °C and −80 to −60 °C respectively; 2–8 °C for 30 days, 2–8 °C for 5 days; room temperature ≤ 12 and ≤2 h. Furthermore, a number of research projects are in clinical trials to assess the efficacy of mRNA vaccines against serious viral infection including HIV, influenza, ZIKA and rabies, etc. [61]. It is concluded that mRNA vaccines will be a major target for the research and vaccine development in future.

### 2.2. Subunit Vaccines

SARS-CoV-2 subunit vaccines comprise distinct viral antigenic particles having strong immunogenicity produced as recombinant proteins [37]. Subunit vaccines are relatively safer, as there is no risk associated with dealing with live virus particles or genome integration during the production procedure. Therefore, subunit vaccines are generally considered to be safe and well-tolerated vaccines. In addition to safety and tolerance, this type of vaccine strategy often has a need for effective adjuvants to achieve a stronger immune response. They are comparatively safer than vaccines having inactivated pathogens. These vaccines either utilize the S protein or receptor-binding domain (RBD) of the S protein as antigens [62]. Examples of subunit vaccine shaving the S protein as an antigen are the SCB-2019 vaccine by Clover Biopharmaceuticals AUS Pty Ltd., Chengdu, Sichuan, China, NVX-CoV2373 by Novavax, Covax-19 by GeneCure Biotechnologies; Vaxine Pty Ltd., Adelaide, Australia and MVC-COV1901 by Medigen Vaccine Biologics Corp. Whereas subunit vaccines with the RDB domain of the S protein as an antigen include KBP-COVID-19 by Kentucky BioProcessing, Inc and the vaccine from Anhui Zhifei Longcom Biologic Pharmacy Co., Ltd., Anhui, China. The subunit vaccine NVX-CoV2373 by Novavax using the saponin-based Matrix-M adjuvant is currently in a phase 3 trial and already authorized by the WHO and FDA to be used in the pandemic with an efficacy rate of 89.5%. The Novavax vaccine is usually administered in two shots comprising 5 µg of protein and 50 µg of Matrix-M adjuvant with a storage life of months at 2–8°C [62]. On the contrary, other subunit vaccines are currently in phase 1 and 2 clinical trials. There are 55 protein subunit vaccines which are currently in the preclinical evaluation stage. In a clinical study, a total of 83 individuals received NVX-CoV2373 with adjuvant and 25 individuals received NVX-CoV2373 without adjuvants. Results demonstrated no or very mild local and systematic reactogenicity with little side effects such as headache, malaise, and fatigue. Adjuvant-based trials of subunit vaccines exhibited elevated immune response and an induced product of Th1 cells. In addition, anti-spike immunoglobulin G (IgG) with a neutralization response was observed with two shots of adjuvant-based subunit vaccine. Note that individuals having symptoms of COVID-19 demonstrated 100 times greater anti-spike IgG levels [62].

### 2.3. Inactivated Virus Vaccines

The second most effective vaccine platform for SARS-CoV-2 is based on inactivated virus by the production of antigenic isotopes. In this approach, virus particles have been cultivated and inactivated chemically to attain stable antigens epitopes conformers. Inactivated vaccines can induce diverse immunologic response as they comprised multiple antigenic components [64,65]. These vaccines are delivered as two doses. Examples of such vaccines are Sinopharm and Sinovac. These vaccines have been authorized by the WHO in for use in COVID-19 patients in China, Brazil, Columbia, Chile, UAE, Peru, Turkey, and Indonesia as emergency use authorization. Two doses are administered 14 days apart for Sinovac Biotech whereby they were 21 days apart for Sinopharm with approximately 80% efficacy. These vaccines are stored at 2–8 °C [57].

### 2.4. Viral Vector Vaccines

The basic principle of viral vector vaccines relies on the genetic sequence of the antigen of interest, which was normally expressed by an engineered replication incompetent virus. The success rate of viral vector vaccines varied with often limited pre-existing immunity to the adenovirus. These type of vaccination approaches have been applied earlier to treat HIV, tuberculosis or malaria [66]. Two SARS-CoV-2 vaccines have shown promising outcomes based on the use of adenoviruses with minimal pre-existing immunity in the United States (US) and Europe (EU) [67]. It is important to note that viral vector vaccines have a high risk of adverse events, as the attenuation is crucial for safety. These vaccines produce stronger immune response by imitating the infection caused by genuine virus [33,34,35,36,37]. These include adenovirus serotype 26 vector vaccine (Ad26.CoV2. S; Johnson & Johnson, New Brunswick, NJ, USA) and chimpanzee adenovirus vector vaccine (ChAdOx; AstraZeneca, Cambridge, UK), which exhibit efficacy against death and hospitalization caused by SARS-CoV-2 [57]. However, the efficacy varies with the disseminated infections and mutations associated with SARS-CoV-2. Doses are given as two jabs and are 28 days apart. Storage conditions involve 2–8 °C for 6 months [57].

### 2.5. Live Attenuated Vaccines

It is known as the classic technique for viral immunization and contains numerous antigenic constituents. Live attenuated vaccines provide effectivity which relies on strong immunogenicity and stimulant Toll-like receptors (TLRs). Previous vaccines based on the similar principle have shown promising and long-lasting protection among smallpox, measles and poliovirus patients and helped to stamp out smallpox successfully [68]. The upgrowth of these vaccines is well established along with the special handling of live virus particles during production process [37]. There are some supplementary requirements concerning safety checks of these vaccines, as they can propagate infections in immunocompetent patients [69]. Live attenuated vaccines exhibit certain benefits which includes rapid development, induction of strong immune response and multivalency. Whereas, there are multiple risk factors associated with live attenuated vaccines such as a higher risk of infection, complications in manufacturing and possibility of reversion [23,70,71,72].

### 2.6. Virus-Like Particles Vaccines

These vaccines involve numerous viral structural proteins, which help to imitate the conformation of viruses. Since these vaccines do not contain any viral genome, they are extremely protected and completely non-infectious. The utilization of a mixture of primary proteins from another virus produces recombinant virus-like particles (VLPs). These immunizations are often produced in plants, as the plant cells are at an ideal stage for the development of oral vaccines. These vaccines are commonly known as edible vaccines. Plants are used as a host to acquaint genes for viral proteins with the use of pathogenic *Agrobacterium*. It results in an infection, which leads to integrating the gene of interest into the chloroplast genome [62]. Consequently, this change prompts the biosynthesis of virus-like particles in plants in a large quantity. The precedent immunization against Avian influenza H5 (AIV) and influenza A virus (A/H1N1, A/H3N2) was well protected and endured. In addition, these vaccines exhibited a favorable response against Newcastle and Lyme diseases. Furthermore, VLPs vaccines were affirmed as areas of strength for cross-reactive humoral and cellular responses. Regarding SARS-CoV-2, there are 15 vaccines in pre-clinical trials. Among them, (RBD SARS-CoV-2) HBsAg VLP and plant-derived VLP (Medicago) were recently enrolled in clinical phase 1 trials. The VLPs vaccine shows more stability at higher temperature, i.e., 25°C in comparison with other SARS-CoV-2 vaccines for 2 weeks [62].

### 2.7. DNA Vaccines

Deoxyribonucleic acid (DNA) vaccines primarily contain plasmid DNA that encodes at least one antigen. The production of antigen proteins occurs by penetration into the nucleus. DNA vaccines also have significant therapeutic potential by enhancing T-cell induction and antibody production. Among other advantages, disposal of the utilization of live virus, manageable freeze-dying, biocompatibility of plasmid DNA, low-cost manufacturing, long shelf life and storage conditions are considered favorable. In addition, there is a possibility of oral application as well [71,72]. Risk factors associated with DNA vaccines are poor immune response, toxicity, and genetic integration [23,70]. Overall, 13 candidates as DNA vaccines are in pre-clinical trials, of which four are under development in phase 1/2 clinical trials, which includes INO-4800 (International Vaccine Institute; Inovio Pharmaceuticals), nCov vaccine (Cadila Healthcare Ltd., Ahmedabad, India), AG0301-COVID19 (AnGes, Inc., Osaka, Japan) and GX-19 (Genexine, Inc., Seongnam-si, Republic of Korea) [23,64,70]. Pre-existing DNA vaccines were in trials for a few decades whereby new advancement in mRNA vaccines has been empowered by mRNA modification, storage and delivery methods. In comparison with DNA vaccines, mRNA entities are smaller, modest and transcription is performed in vitro, whereby DNA molecules must cross the nuclear membrane to be transcribed, and they possess comparatively low immunogenicity. Therefore, they pass over the in vivo transcription procedure yet tend to biodegradation and the investiture of unwanted interferons (INFs) immune responses. A study on DNA vaccines discovered that rhesus macaques were able to develop humoral as well as cellular responses. Furthermore, neutralizing antibodies induced by DNA vaccines were reported to have protective efficacy. It is warranted to conduct further studies to explore whether DNA vaccines are able to induce long-term neutralizing antibodies and the effect of non-neutralizing antibody responses in the related diseases [73] (Table 3).

### 2.8. Mix and Match Vaccine/Boosters

Recently, mix and match vaccine trials have been successfully attempted in animals and humans. According to the animal study, the use of Sputnik V vaccine and AstraZeneca as the first and second dose in mice led to enhanced immunity without having any concerned problems [63]. In another animal study, a similar study was conducted in mice with the use of self-amplifying RNA vaccine (saRNA) and adenovirus carrier vaccine (ChAdOx1 nCoV-19/AZD1222) as the first and second dose. As a result, higher antibody response was observed with mix and match vaccines as compared to single vaccine doses [74]. saRNA are a kind of mRNA vaccine which encodes a replicase sequence which plays a crucial role in reduced injection dose and the number of required injections by improved mRNA half-life [62]. Following the successful results of mixing SARS-CoV-2 vaccines in animals, a number of studies were conducted in humans using both shots of different vaccines, which was found to be effective in triggering immune response as compared to a single shot of either vaccine [75,76,77,78]. In a study by Barros-Martins and team, individuals received a dose of AstraZeneca vaccine as the first dose followed by Pfizer vaccine as the second dose. As a result, the observed immune response of IgG, Immunoglobulin A (IgA), and anti-spike (S) was 11.5 times superior along with better humoral immune response to those injecting with both doses of AstraZeneca [79]. A study by Liu and co-workers showed that for people jabbed with ChAdOx1 nCoV-19 vaccine as the first dose and mRNA vaccine (BNT162b2 or mRNA-1273) as the second dose, their spike-specific IgG, neutralizing antibodies, spike-specific CD8 T-cell levels, spike-specific CD4 T cells, and humoral and cellular immune responses were potentially enhanced [21]. A Spanish study is currently in phase 2 trials utilizing a first dose of Oxford AstraZeneca and second dose of Pfizer BioNTech, which resulted in an elevation of antibody levels by 150 times more after 14 days of the second dose in comparison with a control group that received only AstraZeneca as the first dose [77]. A study conducted by Oxford University reported stronger immune response with mixing Oxford AstraZeneca and the Pfizer BioNTech vaccine. However, Com-COV trials also reported that a heterologous vaccine could induce more side effects when compared with both doses of a single vaccine [21,76]. Different possible mechanisms have been implied for the cause of a higher immune response in the use of mixing two different vaccines. For instance, it is suggested that a diverse constitution of vaccines elicits different projections of the immune system, hence resulting in prolonged combined cellular and humoral immunity. Heterologous vaccines can help to evoke humoral immunity in different ways by attaining higher IgG levels and neutralizing antibodies. However, the exact mechanism behind the enhanced immune response associated with the heterologous vaccine is not completely understood. It is clearly evident from the studies that mixing SARS-CoV-2 vaccines provides potentially enhanced IgG antibodies, neutralizing antibodies, and cellular immune response as compared to homologous SARS-CoV-2 vaccination [80]. This recommends a similar mechanism as previous other heterologous vaccines for the underlying mechanism in the SARS-CoV-2 heterologous vaccine for promoting higher immune response [81].

At present, bivalent mRNA vaccines are recommended by the Center for Disease Control and Prevention (CDC) to use as a booster at least two months after the last vaccine. Monovalent mRNA vaccines are no longer in use as a booster dose. These bivalent booster doses include Original and Omicron BA.4/BA.5 developed by Pfizer-BioNTech and Original and Omicron BA.4/BA.5 developed by Moderna, which were recently authorized by the FDA [82].

### 2.9. Vaccines in Clinical Trials Targeting SARS-CoV-2

Multiple vaccine candidates were being investigated since the outbreak of COVID-19. Many of them were approved and provided to humans to reduce the burden of highly contagious SARS-CoV-2, which include mRNA, DNA, subunit, inactivated virus and so on. As of June 2022, 163 different vaccines candidates are being tested in pre-clinical and clinical trials such as protein subunit, viral vector, VLP, nucleoside unmodified and modified linear nucleoside RNA lipid nanoparticle (LNP) LNP-mRNA, circular RNA (cirRNA), saRNA and DNA vaccines. Out of the 163, 146 candidates are being confirmed by several companies, whereby 7 of them are no longer under clinical development considerations [83,84]. LNP-mRNA vaccines from BioNTech/Pfizer named BNT162b1 and BNT162b3 are evaluating in phase 3 and 2 clinical trials, respectively. BNT162b1 demonstrated positive clinical outcomes and was selected for a pivotal efficacy study; however, the clinical outcomes pertaining to BNT162b3 have not been published yet. Likewise, mRNA-1273.211 and mRNA-1273.351 developed by Moderna are being examined at clinical phase 3 and 2 and elevation in neutralization titer variant is detected specifically by mRNA-1273.211. TAK-919 from Takeda/Moderna and ChulaCov19 from Chulalongkorn University are in ongoing phase 2 clinical trial, and results are yet to be published [83,85,86,87]. Several unmodified mRNA vaccines against SARS-CoV-2 were investigated at early stages of clinical study which includes MRT5500 from Bio/Sanofi, SW-0123 from Stemirna Therapeutics/Shanghai East Hospital and EG-COVID from eyeGENE, and findings are still not known. Whereas a bunch of unmodified mRNA vaccines including CVnCoV from CureVac, ARCoV from AMS/Walvax/Suzhou, PTX-COID19B from Providence Therapeutics, and SD5670 from Daiichii Sankyo showed efficacy without any safety concerns [88,89,90]. There are two saRNA vaccines under clinical trials 2 and 3, namely LNP-nCOV saRNA (Imperial College London) and ARCT 021, ARCT 154, ARCT 165 (Arcturus Therapeutics/Duke-NUD Medical school) and exhibited dose-dependent responses against SARS-CoV-2. Some other saRNA vaccine candidates from GlaxoSmithKline (GSK), Imperial College London/MRC/UVRI/LSHTM/Uganda Research Unit, and VLP Therapeutics Japan are still in early stages of clinical study and have not recorded any result yet [83,90]. FINALAY-Fr-1A is a subunit vaccine based on a recombinant protein antigen, a dimer of RBD developed by Soberna Plus, Finlay Vaccine Institute and the Centre of Molecular Immunology, Havana, Cuba. A study conducted by Ochoa-Azze and co-workers reported the significant efficacy of FINLAY-Fr-1A in phase 2 a and b of the clinical study without any adverse side effects. The study was conducted on COVID-19 convalescent participants and proved to enhance humoral immunity in participants, which suggests a prevention of severe reinfection by SARS-CoV-2 variants of concern [71,91].

Recently, BNT162b2 and mRNA-1273 vaccines from BioNTech and Moderna are under clinical development phases targeting specific population such as immunocompromised patients and young children aged between 6 months and 18 years [83].

## 3. Challenges Associated with SARS-CoV-2 Vaccines

Numerous aspects must be considered when there is a discussion on challenges associated with SARS-CoV-2 vaccines. Among other options, vaccination is known as the most effective way to prevent many infectious diseases. Vaccines work by inducing immune response by a human body as infection occurs naturally [92]. Vaccine requires a foreign agent which works as vaccine-active agent and induces immune memory to have successful expression with pathogens [93]. There is a resistance and shortage in the availability of COVID-19 vaccines especially in poor regions and with the emergence of new variants. Because of the aforementioned challenges, many countries were forced to mix SARS-CoV-2 vaccines [74,94], which led to a significant increase in IgG and neutralizing antibodies with strong cellular immune response [80,95]. Moreover, heterologous SARS-CoV-2 vaccines have resulted in higher neutralizing antibody levels in comparison with homologous vaccines [80]. Consequently, there is a huge utilization of mixing SARS-CoV-2 vaccines in both developing and industrialized countries. This new strategy started with a hope to immunize a prominent percentage of their population with more efficacy against SARS-CoV-2 without adverse side effects [96]. This will help the regions which are affected by a shortage of SARS-CoV-2 vaccines with more effectivity against infections linked to SARS-CoV-2.

### Viral Mutations

Multiple variants of SARS-CoV-2 have been appeared across the entire world since 2019. There are greater chances to increase the transmission of viruses mainly when infections are still huge [97]. It is noteworthy to mention that the higher rate of infections is directly proportional to an increase in mutations and thus helping the virus in survival and proliferation [98]. The herd immunity and crowd resistance are the factors directly involved with virus evolution [99]. These genetic changes in the human genome associated with virus are essential because of certain elements such as shortfall of immunity against new pathogen. Furthermore, the transformation paces of SARS-CoV-2 that encode a protein with editing capability increase the speed of replication processes [100]. A study was conducted to examine upcoming mutations of SARS-CoV-2 by studying the gene sequencing of SARS-CoV-2. The outcomes showed that 26,844 single transformations were continued in 203,346 human genomes of SARS-CoV-2, while the most well-known changes included S proteins and non-structural protein 3 (NSP3) within the period of three years from 2020 to 2022 [101]. Around 5000 transformations were perceived in the S protein by the end of December 2020. New strains emerged from all over the world, particularly in United Kingdom, e.g., lineage B.1.1.7 and variant 20I/501Y.V1. These variations are caused by several spike (S) protein changes, including deletion 145, N501Y, deletion 69–70, D614G, A570D, T716I, P681H, D1118H, and S982A [101]. Approximately 82 countries reported this variant with a higher rate of transmission, disease severity and reduced neutralization efficiency [90]. Furthermore, another variant was discovered in South Africa lineage B.1.351, variant 20H/501Y.V2) with the involvement of eight mutations through S protein: D80A, L18F, R246I, D215G, E484K, K417N, A701V, and N501Y. This variant was related to elevated reinfections and transmissions and was reported in 35 countries [82,102]. The variant reported from Brazil has lineage 20 J/501Y.V3 and lineage P.1, mutated through three S protein mutations, N501Y, E484K, and K417N, in common with 20J/501Y.V2 [100]. The variant was reported in 14 countries with the same risk factors as the Brazil variant. The important factor of all these variants is that they share N501Y mutation concerning the SARS-CoV-2 spike (S) protein which is considered as the essential target of most vaccines [103]. An additional variant, Omicron, was discovered in South Africa in November 2021. The mutations linked with this variant are B 1.1.529 lineage and H655Y, N679K, P681H, R203K, G204R, E484, D614G, R93K, and G204R. It was reported in 87 countries with higher reinfection and transmission rates [102]. The first case of the novel variant known as Delta was reported in India in October 2020 having B 1.1.1617 lineage. The Delta variant demonstrated higher infections and transmission abilities [104]. After the Delta variant, there is a new ongoing wave of SARS-CoV-2 virus caused by Omicron and its subvariants. Omicron is reported to have the highest number of mutations in comparison with other variants. Omicron and its subvariants showed a relative lower severity of infection than that of other variants, but the transmission rate is elevated and reported as almost 3.8 times that of the Delta variant [105]. The most concerning insight of the Omicron variant is the ability to escape the immune system and consequently reduce the efficacy of vaccines. To date, more than 50 mutations have been recorded of the Omicron variant, 32 of which are in the spike protein of SARS-CoV-2 [106]. There is limited research available on the effects particularly regarding the transmission and neutralization of new variants on current SARS-CoV-2 vaccines; therefore, extensive and deep studies are warranted in the future.

## 4. Conclusions

The causative agent of the current pandemic is SARS-CoV-2, which is still causing mortalities on a higher scale. There is still a huge socio-economic burden on healthcare services throughout the world by introducing various methods to control the viral infection. The exact nature and ability of the SARS-CoV-2 virus regarding transformation is still vivid after three waves of the pandemic in a row. However, many reports about the new epidemic are contrasting. The first-generation SARS-CoV-2 vaccines targeting spike glycoprotein have showed potential in reducing its transmission. Unfortunately, the effectiveness of the currently available vaccinations was compromised by locally circulating variants in the disease. To assess the durability of protective antibodies against new variations, therefore, a long-term study of neutralizing activity is required. Moreover, further study is necessary to properly comprehend the correlation between SARS-CoV-2 vaccinations and uncommon adverse effects as well as long-term studies to evaluate delayed responses after vaccination (Figure 3).

## Figures and Tables

**Figure 1 diseases-11-00064-f001:**
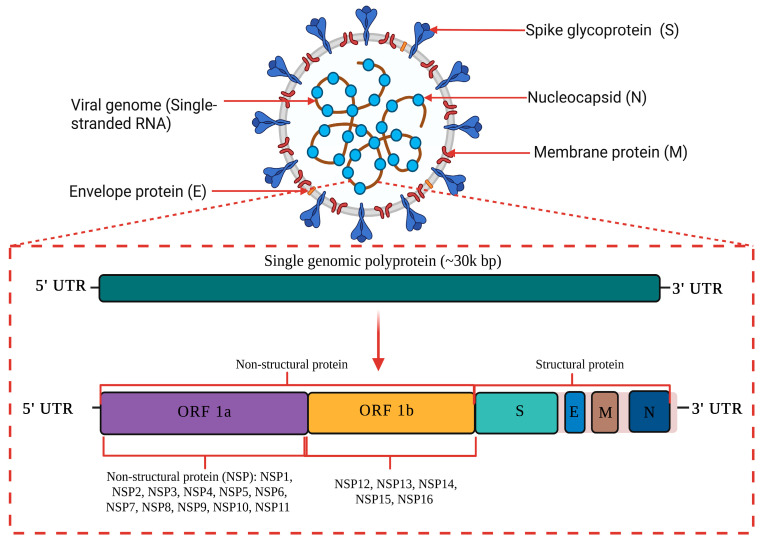
A schematic diagram of SARS-CoV-2 genome structure. SARS-CoV-2 is an enveloped RNA virus with an approximately 30 kilo base pair (kbp) genome size. The primary structural proteins of SARS-CoV-2 are membrane (M), spike (S) and nucleocapsid (N) proteins. The genome of SARS-CoV-2 consists of 14 open reading frames (ORFs) which encode for 27 structural and non-structural proteins. This image was created with Biorender.com.

**Figure 2 diseases-11-00064-f002:**
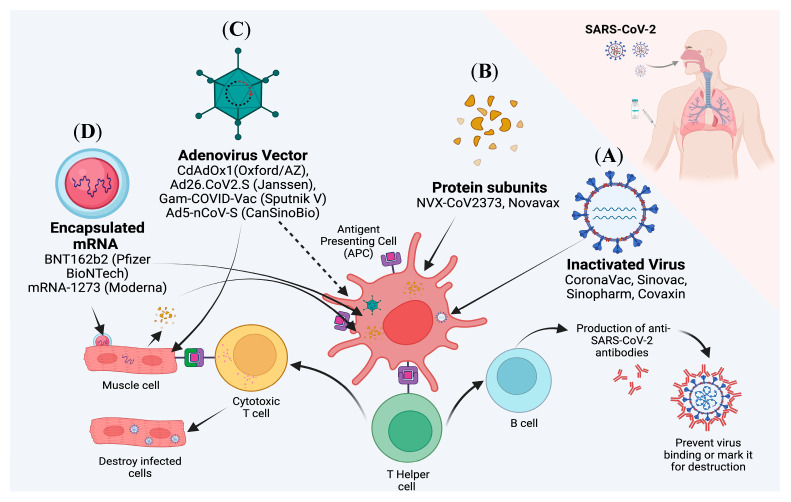
A schematic diagram of current active development of SARS-CoV-2 vaccines as of September 2022. There are several types of vaccines which include (**A**) Inactivated viral vaccines, (**B**) Protein subunits vaccines, (**C**) Adenoviral vector vaccines and (**D**) mRNA vaccines. Aside from the whole inactivated virus in inactivated viral vaccines, other vaccines targeted the binding domain of SARS-CoV-2 Spike (S) protein. Moreover, the vector and mRNA vaccines targeted muscle cells at the injection site. The muscle cells produced some portion of the SARS-CoV-2 S protein, which is then presented by MHC Class I to antigen-presenting cells (APCs) and cytotoxic T cells. In contrast, inactivated viral vaccines and protein subunits vaccines are directly taken up by APCs. The APCs then presented the S protein antigen to T helper cells as well as B cells, which in turn activated sequential activities of humoral and cellular immune response against the SARS-CoV-2 S protein. These immunological activities also indirectly generate memory T cells and B cells for protection against future exposure against the virus. This image was created with Biorender.com.

**Figure 3 diseases-11-00064-f003:**
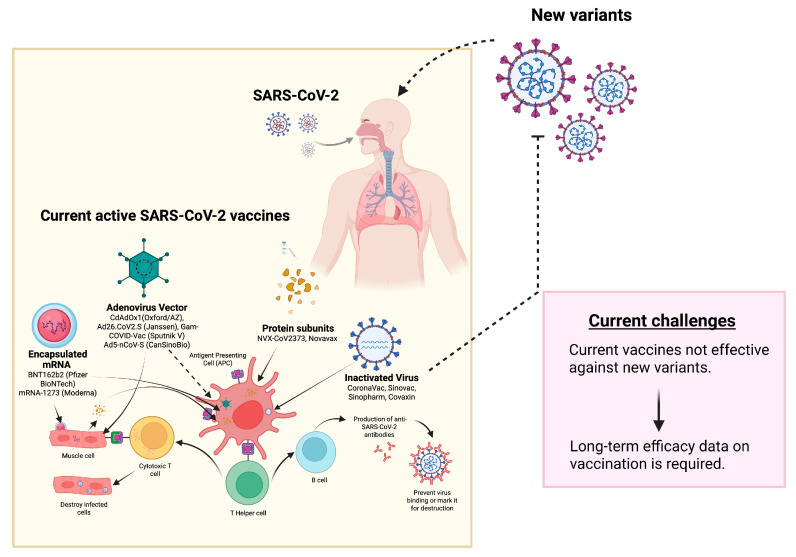
Current perspective of vaccine development against SARS-CoV-2. The figure was made with Biorender.com (accessed: 30 December 2022).

**Table 2 diseases-11-00064-t002:** Omicron subvariants under monitoring (as of 9 February 2023).

Pango Lineage #. (+Mutation)	GISAID Clade	Next Strain Clade	Relationship to Circulating VOC Lineages	Spike Genetic Features	Earliest Documented Samples	Pathophysiology	Epidemiology
BF.7 *	GRA	22B	BA.5 sublineages	BA.5 + S:R346T	24 January 2022	- first detected 13 May 2022 in Belgium - spike protein mutations (at amino acid R346T) leading to their ability to escape neutralizing antibody [46] - R346T may contribute to reduced sensitivity to mAbs, increased binding to (ACE2) receptor, and increased expression of the RBD [47,48,49]	- As of 4 October 2022, 9809 sequences of the BF.7 sublineage have been reported globally and are mostly prevalent in Europe [46]
BQ.1 ^$^	GRA	22E	BA.5 sublineages	BQ.1 and BQ.1.1: BA.5 + S:R346T, S:K444T, S:N460K	7 February 2022	- currently, there are no epidemiologic data that show an increase in disease severity	- has a prevalence of 6% and has been detected in 65 countries [50] - no data on severity or immune escape from studies in humans
BA.2.75 ^§^	GRA	22D	BA.2 sublineage	BA.2.75: BA.2 + S:K147E, S:W152R, S:F157L, S:I210V, S:G257S, S:D339H, S:G446S, S:N460K, S:Q493R reversion	31 December 2021	- additional mutations may be capable of escaping neutralizing antibodies [51]	- first detected in India in May 2022 [52]
CH.1.1 ^§^	GRA	22D	BA.2 sublineage	BA.2.75 + S:L452R, S:F486S	27 July 2022	- mutation in the L452R substitution in the spike protein (previously discovered in the Delta and Omicron BA.4/5 variants) [53] - lower infectivity, highly resistance to neutralization by bivalent sera, lower efficiency of syncytia formation [54]	- as per reported by CDC, CH1.1 is the 5th most spreading variants, which accounted for 1.6% of total new cases in the U.S. [54] - as of January 15, it was detected in the U.K., where it accounted for 28.2% of new sequence infections compared to XBB.1.5′s 10.9% [54]
XBB ^µ^	GRAA	22F	Recombinant of BA.2.10.1 and BA.2.75 sublineages, i.e., BJ1 and BM.1.1.1, with a breakpoint in S1	BA.2+ S:V83A, S:Y144-, S:H146Q, S:Q183E, S:V213E, S:G252V, S:G339H, S:R346T, S:L368I, S:V445P, S:G446S, S:N460K, S:F486S, S:F490S	13 August 2022	- early evidence pointing at a higher reinfection risk as compared to other circulating Omicron sublineages - currently, no available data to support escape from recent immune responses induced by other Omicron lineages [55] - further studies are needed, the current data do not suggest there are substantial differences in disease severity for XBB * infections	- has a global prevalence of 1.3% and it has been detected in 35 countries [55]
XBB.1.5	GRA	23A	Recombinant of BA.2.10.1 and BA.2.75 sublineages, i.e., BJ1 and BM.1.1.1, with a breakpoint in S1	XBB + S:F486P	15 January 2022	- further studies are needed; the current data do not suggest there are substantial differences in disease severity for XBB * infections	- as of January 15, it was detected in the U.K., where it accounted for 10.9% of new sequence infections [54]
XBF	GRA		Recombinant of BA.5.2.3 and CJ.1 (BA.2.75.3 sublineage)	BA.5 + S:K147E, S:W152R, S:F157L, S:I210V, S:G257S, S:G339H, S:R346T, S:G446S, S:N460K, S:F486P, S:F490S	27 July 2022	- further studies are needed; the current data do not suggest there are substantial differences in disease severity for XBB * infections	- responsible for the recent surge of COVID cases in Australia [55] - first case detected in the Philippines on 28 January 2023 (specimens collected in December 2022) [56] * very little to no information on XBF, mainly reported on online newspaper only.

# includes descendent lineages; * additional mutations outside of the spike protein: N: G30, S33F, ORF9b: M26, A29I, V30L; ^$^ additional mutation outside the spike protein: ORF1a: Q556K, L3829F, ORF1b: Y264H, M1156I, N1191S, N: E136D, ORF9b: P10F; ^§^ additional mutations outside of the spike protein: ORF1a: S1221L, P1640S, N4060S, ORF1b: G662S, E: T11A; ^µ^ additional mutations outside of the spike protein: ORF1a: K47R, ORF1b: G662S, S959P, E: T11A, ORF8: G8*.

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
