# Peer review of "Development of SARS-CoV-2 Vaccine: Challenges and Prospects"

_diseases, 2023, doi:10.3390/diseases11020064_

Round 1

Reviewer 1 Report

This review provides an overview of the pathogenicity of the new coronavirus infection (COVID-19) and current preventive and therapeutic measures. In particular, the development of vaccines against COVID-19 is explained in detail, and the content is timely. It is also of interest not only to specialists, but also to readers from a variety of disciplines.

[Major point]

Nothing

[Minor points]

1. Line 81: Align sentences to the left!

2. Line 192: Change “SARS-_CoV-2” to “SARS-CoV-2”. Closing the space between “SARS-” and “CoV-2”.

3. Table 1: “Country” is split into “Count” and “ry”. Stick them.

4. Table 1: Dose/Dosage of “Ad26.CoV2.S” is “5 x 1010viral particle”. Put a space between "5 x_1010" and "viral particle".

5. Table 1: Since there is a space at the head of ``Inactivated virus'' in the “Type'' column of ”BBIBP-CorV'', move it to the left.

6. Figure 2: Insert (A)-(D) in the figure. They might be missing.

7. Line 334: Change “SARS-_CoV-2” to “SARS-CoV-2”. Closing the space between “SARS-” and “CoV-2”.

8. Line 362: Close the space between "Toll-" and "like".

9. Line 367: Isn't "It" a mistake for "it"?

10. Line 388: Isn't "Amon" a mistake for "Among"?

11. Line 527: Change “SARS-_CoV-2” to “SARS-CoV-2”. Closing the space between “SARS-” and “CoV-2”.

12. Lines 527-530: The meaning this sentence is unclear. Do you need “human genomes” in the sentence?

13. Line 534: Need period after "[89]".

14. Line 537: Isn't "E484" a mistake for "E484K"?

Author Response

1.Line 81: Align sentences to the left!

Done

2. Line 192: Change “SARS-_CoV-2” to “SARS-CoV-2”. Closing the space between “SARS-” and “CoV-2”.

Done

3. Table 1: “Country” is split into “Count” and “ry”. Stick them.

Done

4. Table 1: Dose/Dosage of “Ad26.CoV2.S” is “5 x 1010viral particle”. Put a space between "5 x_1010" and "viral particle".

Done in table 1.2.

5. Table 1: Since there is a space at the head of ``Inactivated virus'' in the “Type'' column of ”BBIBP-CorV'', move it to the left.

Done

6. Figure 2: Insert (A)-(D) in the figure. They might be missing.

Done

7. Line 334: Change “SARS-_CoV-2” to “SARS-CoV-2”. Closing the space between “SARS-” and “CoV-2”.

Done

8. Line 362: Close the space between "Toll-" and "like".

Done

9. Line 367: Isn't "It" a mistake for "it"?

Corrected

10. Line 388: Isn't "Amon" a mistake for "Among"?

Corrected

11. Line 527: Change “SARS-_CoV-2” to “SARS-CoV-2”. Closing the space between “SARS-” and “CoV-2”.

Done

12. Lines 527-530: The meaning this sentence is unclear. Do you need “human genomes” in the sentence?

Corrected as suggested, Line 525

13. Line 534: Need period after "[89]".

Added as suggested, Line 532

14. Line 537: Isn't "E484" a mistake for "E484K"?

It is E484K mutation.

Reviewer 2 Report

  • Major comments: 

In this review, Tooba Mahboob et al. briefly summarized SARS-CoV-2 pathophysiology, epidemiology, mutant variants, and management strategies, and discussed SARS-CoV-2 vaccines, including mRNA vaccines, DNA vaccines, subunit vaccines, inactivated virus vaccines, viral vector vaccine, live attenuated virus vaccines and mix and match vaccines based on their mechanism, administration, storage, stability, safety, and efficacy.

The review is relevant to the field, and the finding of the study may be helpful to the research community and general audience.

  • General concept comments

Here are some considerations for the review:

a.              The table showing the circulating VOCs and Omicron subvariants under monitoring is missing.

b.              Please list a table to illustrate the characteristics like pathophysiology and epidemiology of the circulating VOCs and Omicron subvariants.

c.              Table 1(SARS-CoV-2 Vaccines) was not completed or comprehensive, according to the description of the section 2. Current update on various vaccines targeting SARS-CoV-2 of the review.

d.              Also, for Table 1, please provide references for each vaccine provided, especially the Overall efficacy data.

e.              Since the Omicron and its subvariants are predominant now, please summarize the effects of new variants on current SARS-CoV-2 vaccines, as there is already much literature about them.

The review could be further improved by including the following suggestions or considerations listed in specific comments. 

  • Specific comments:
  1. The grammar of the whole text should be thoroughly checked.
  2. Line 55, “SARS-CoV” is not the same as “SARS-CoV-2”.
  3. Line 57, “six species”? It should be seven species now.
  4. Lines 179-180, the description here, is outdated.
  5. Lines 193-194, IL-6 approved by FDA for COVID-19 patients?
  6. Lines 288-289, the description here may be outdated.
  7. Line 301, missing the unit information of the number “12” here.
  8. Line 320, it seems like NVX-CoV2373 was also approved by FDA under EUA.
  9. Line 332, the number “51” here is misleading.
  10. Lines 404-416, It seems like the description here is about mRNA vaccine rather than DNA vaccine.
  11. Line 422, please detail the self‐amplifying RNA vaccine (saRNA).
  12. Line 495, the references here are misleading.
  13. Lines 546-547, the information on the mutations linked with B 1.1.529 variant is not complete and misleading.

Round 2

Reviewer 2 Report

I think that the manuscript has been improved, and the authors have addressed most of my concerns.

Please check the whole text thoroughly and fix the problem of “[58 k-79]” in line 821.

Author Response

Thank you for constructive comments. Entire manuscript has been checked thoroughly as suggested.